# Physicochemical Properties and Storage Stability of Food Protein-Stabilized Nanoemulsions

**DOI:** 10.3390/nano9010025

**Published:** 2018-12-25

**Authors:** Yangyang Li, Hua Jin, Xiaotong Sun, Jingying Sun, Chang Liu, Chunhong Liu, Jing Xu

**Affiliations:** College of Science, Northeast Agricultural University, Harbin 150030, Heilongjiang, China; ccwasdrdxnlyy@126.com (Y.L.); jinhua@neau.edu.cn (H.J.); sunxt0523@163.com (X.S.); asunjinying@126.com (J.S.); aaxuchang@126.com (C.L.); liuchunhong@neau.edu.cn (C.L.)

**Keywords:** ultrasound, proteins, nanoemulsions, physicochemical properties, storage stability

## Abstract

This study investigated the preparation and properties of corn oil nanoemulsions stabilized by peanut protein isolate (PPI), rice bran protein isolate (RBPI), soybean protein isolate (SPI), and whey protein isolate (WPI). The mean droplet diameter of four protein-stabilized nanoemulsions prepared via ultrasound method was less than 245 nm. PPI-stabilized nanoemulsions showed better stability for 4 weeks, while the mean droplet diameter of RBPI-stabilized nanoemulsions had exceeded 1000 nm during the third week of storage. Fourier transform infrared and interfacial tension (IT) analysis showed that the higher level of disordered structure and lower IT of proteins made the stability of nanoemulsions better. Moreover, bivariate correlation analysis discovered that α-helix (*p* < 0.01) and β-turn (*p* < 0.05) of proteins were related to the mean droplet diameter of nanoemulsions, the random coil (*p* < 0.05) was related to the zeta potential of nanoemulsions. This study provided new idea for the relationship between the structure of protein and properties of protein-stabilized nanoemulsions.

## 1. Introduction

Nanoemulsions are composed by minute oil droplets (average droplet diameter less than 500 nm) which are maintained in the aqueous phase. Each nanoemulsion oil droplet is usually stabilized by emulsifier molecules. Unlike most traditional emulsions, nanoemulsions can resist gravitational settling or coalescence more effectively, and they are overall more stable as they possess larger surface area of droplets [1]. Moreover, nanoemulsions can be designed to have different rheological, optical, and stability features via control of their structures and compositions [2,3,4]. Therefore, nanoemulsions have acted as an excellent system for the encapsulation and delivery of bioactive compounds which are lipophilic in nature.

The nanoemulsions characteristics, such as the droplet size distribution, optical and rheological stability, are governed by emulsification technologies and emulsifier types [5]. High and low energy emulsification approaches are usually employed for nanoemulsions preparation. Spontaneous emulsification (microemulsion dilution and solvent diffusion) and phase transition (by adjusting component or temperature) are two commonly used low energy emulsification approaches [6]. High energy emulsification approaches include high pressure homogenization, microfluidization and ultrasound, which can generate strong destructive forces and disrupt large droplets into smaller ones, thereby mixing the oil phase and the water phase. Compared with low energy emulsification methods, high energy approaches are more frequently utilized to produce nanoemulsions, especially in large scale production of food industry. The ultrasonic cavitation can increase diffusion rates and disperse aggregates in the formation of nanoemulsions [7,8,9]. As determined in previous studies, when producing nanoemulsions with a given droplet size, the efficiency of ultrasound-based approach is 18 times higher than that of microfluidization [10]. Additionally, ultrasonic techniques are lower cost, requiring less maintenance and handling time than other mechanical approaches. At present, ultrasonic approaches are widely used for nanoemulsions preparation [11,12,13,14,15].

At the beginning, several synthetic surfactants (including Tweens and Spans) were used as high performance emulsifiers to prepare nanoemulsions. However, considering that the safety of emulsifiers is of crucial importance in food products, the potential toxicity of synthetic surfactants limits their applications in food industry [16,17]. So now there has been a growing attention to the use of food-grade emulsifiers. In recent years, studies have uncovered that food proteins have great potential as safe stabilizers for nanoemulsions [18]. The earliest food proteins as stabilizers for nanoemulsions are animal proteins, such as milk and egg proteins. Subsequently, vegetable proteins, such as those from soy beans and peas, have also become prevailing, because of their good emulsifying ability [19,20,21,22].

Peanut protein and rice bran protein have not only high nutritional value, but also their functional characteristics which have attracted people’s attention [23]. Zhang and Lu (2015) found that peanut protein adsorbed on the oil/water interface in emulsions possessed higher emulsification [24]. Zhang et al. (2017) have demonstrated that ultrasound treatment could unfold the conformation of rice bran protein, exposing its interior functional groups, which were related to its emulsifying property [25]. However, the efficacies of ultrasound-mediated preparation of nanoemulsions from peanut protein and rice bran protein remain largely unknown.

The objectives of our work were to evaluate the properties of ultrasound-mediated corn oil nanoemulsions with peanut protein isolate (PPI) and rice bran protein isolate (RBPI) as stabilizers. During the research, soybean protein isolate (SPI, vegetable proteins) and whey protein isolate (WPI, animal proteins) were used as reference. The secondary structure and interfacial tension of PPI, RBPI, SPI, and WPI were evaluated. And the relationship between the secondary structure of proteins and the forming ability of proteins-stabilized nanoemulsions was analyzed by bivariate correlation analysis. The study provided new information for enhancing the comprehension of peanut protein isolate and rice bran protein isolate as emulsifiers to fabricate food-grade nanoemulsions. Simultaneously, the research provided elementary idea for the mechanism of protein emulsifiers stabilizing nanoemulsions.

## 2. Materials and Methods

### 2.1. Materials

Protein isolate of peanut, rice bran, soybean, and whey were supplied from Harbin High-Tech Co. (Harbin, China). Corn oil was obtained from a local grocery. Nile red and Nile blue were from Sigma-Aldrich (St. Louis, MO, USA). Butylated hydroxytoluene (BHT) was purchased from Merck Co. (Darmstadt, Germany). Trichloroacetic acid (TCA) was from Samchun Pure Chemicals Co. Ltd. (Seoul, Korea). Analytical-grade reagents were used unless otherwise stated.

### 2.2. Nanoemulsions Preparation

Corn oil (1–20% *v*/*v*, respectively) and aqueous phase (10 mM phosphate buffer, pH 7.0) containing peanut protein isolate (PPI) (1–8% *w*/*v*), rice bran protein isolate (RBPI) (1–4% *w*/*v*), SPI (1–7% *w*/*v*), or WPI (1–10% *w*/*v*) were coarsely mixed by a benchtop stirrer. To prevent microbial growth, sodium azide (0.004%, *w*/*v*) was added. The mixtures of corn oil and protein dispersions were further blended via homogeniser (FJ200-SH, Shanghai specimen model factory, Shanghai, China) at 10,000 rpm for 4 minutes to obtain coarse emulsions. Then, an ultrasonic processor (Ningbo Xinzhi Biotechnology Co. Ltd., Ningbo, China) with a 0.636 cm diameter titanium probe immediately processed 20 mL of the coarse emulsions at a determined ultrasonic power (100–500 W, corresponding to the amplitude of 10–50%) and time (5–25 min, pulse duration of on-time 2 s and off-time 3 s). In the process of nanoemulsions preparation, the temperature was controlled at 25 °C by means of ice-water bath.

### 2.3. Droplet Size, Zeta Potential and Apparent Viscosity Measurements

The mean droplet diameter (MDD) of nanoemulsions was determined using a dynamic light scattering (Zetasizer Nano-ZS90, Malvern Instruments, Worcestershire, UK). The particle size results were reported as the Z-average mean diameter calculated from the particle size distribution. To avoid multiple light scattering effects, nanoemulsions were diluted 100-fold with 10 mM phosphate buffer (pH 7.0) and agitated well at 25 °C. The refractive index values used for oil (corn oil) and dispersant (phosphate buffer) were 1.47 and 1.33, respectively.

The zeta potential (ZP) of nanoemulsions systems was investigated using an electrophoresis (Zetasizer Nano-ZS90, Malvern Instruments, Worcestershire, UK). The nanoemulsions were diluted 100-fold with 10 mM phosphate buffer (pH 7.0) and agitated well at 25 °C prior to measurement so as to avoid multiple scattering effects.

The apparent viscosity of nanoemulsions was measured based on shear rate (0.01–100 s^−1^) via a rheometer (DHR-1, TA Instruments, New Castle, PA, USA) in rotational mode. A 40 mm acrylic parallel plate with a 500 μm geometric gap was slowly placed on the sample and equilibrated for 30 s, and then shear was applied for measurement at 25 °C.

### 2.4. Confocal Laser Scanning Microscopy

The dyed nanoemulsions were prepared using Nile red dye (0.1 mg/mL isopropanol) in the oil phase and Nile blue dye (1 mg/mL isopropanol) in the aqueous phase. The dyed nanoemulsions were inspected on a slide with a cover slip using the 63× oil immersion objective lens of a confocal laser scanning microscope (CLSM) (Nikon C2, Nikon lnc., Mississauga, ON, Canada). The dyes were excited using 488 and 633 nm lasers.

### 2.5. Storage Stability

For testing storage stability, the PPI, RBPI, SPI, and WPI-stabilized nanoemulsions were fabricated under the circumstance of the different protein concentrations of PPI (2% *w*/*v*), RBPI (2% *w*/*v*), SPI (3% *w*/*v*) and WPI (4% *w*/*v*), corn oil (3%, 2%, 2%, 3% *v*/*v*, respectively), ultrasonic power 500 W and time 20 min. The influence of reserve time on nanoemulsions was evaluated under the condition of 4 °C for 4 weeks.

#### 2.5.1. Physical Stability Measurement

The physical stability of nanoemulsions was assessed by determining particle size changes per 7 days of storage.

#### 2.5.2. Oxidative Stability Measurement

To assess secondary oxidation products, thiobarbituric acid-reactive substances (TBARS) were measured as previously described by [26]. TBARS levels were established according to 1,1,3,3-tetraethoxypropane standard curve.

### 2.6. Fourier Transform Infrared (FTIR) Spectroscopy of Protein

Infrared spectra of the four different protein samples (PPI, RBPI, SPI, and WPI) were obtained with Bruker Vertex 70 FTIR spectrometer (Bruker Optics, Ettlingen, Germany) via wavenumber ranging from 4000 to 400 cm^−1^. All datas were collected with 4 cm^−1^ window and were the average results of 64 scans. Analyze the secondary structure of four different protein using the software “Peakfit Version 4.12” and the “Gaussian peak fitting” algorithm [27].

### 2.7. Interfacial Tension (IT) Measurement of Protein

The interfacial tension at different concentrations of PPI, RBPI, SPI, and WPI was determined against corn oil using a Du Nouy ring tensiometer (TP681, Timepower, Co., Ltd. Beijing, China) at 25 °C.

### 2.8. Statistical Analysis

All experiments were conducted in triplicate. Results were expressed as mean ± standard deviation. Significance of difference between the means was identified through the Duncan’s multiple-range tests (*p* < 0.05) with SPSS 20.0 software (New York, NY, USA). Pearson’s correlation analysis was conducted to determine the coefficients reflecting the relationship between the secondary structure of proteins and the forming ability of proteins-stabilized nanoemulsions.

## 3. Results and Discussion

### 3.1. Effects of Protein Concentrations on Nanoemulsions MDD and ZP

Effects of different protein concentrations of PPI (1–8% *w*/*v*), RBPI (1–4% *w*/*v*), SPI (1–7% *w*/*v*), and WPI (1–10 % *w*/*v*) on nanoeumlsions MDD were measured (Figure 1a). When the concentration of the emulsifier was low (PPI < 2% *w*/*v*, RBPI < 2% *w*/*v*, SPI < 3% *w*/*v* and WPI < 4% *w*/*v*), the mean droplet diameter decreased significantly (*p* < 0.05) as the concentration increased, because more emulsifier molecules were available to cover more oil–water interfacial areas during the preparation of nanoemulsions. When the emulsifier concentration became higher, the mean droplet diameter remained properly constant, as there were enough emulsifiers covering all the newly formed oil droplets surfaces, thereby the particle size of nanoemulsions no longer continued to decrease [28,29]. Moreover, from Figure 1a, it should be noticed that there was followed by another increase trend of MDD when the concentration of protein further increased to a certain degree. This phenomenon was attributed to the fact that the numbers of interfacial sites were insufficient for the large number of protein molecules and unabsorbed protein molecules led to the aggregation in aqueous phase [30]. The protein concentration required to produce droplets with smallest MDD was 2% for PPI, 2% for RBPI, 3% for SPI, and 4% for WPI, the corresponding MDD was 251.13 ± 5.35, 294.60 ± 7.83, 256.27 ± 8.41 and 266.50 ± 10.57 nm respectively. The results indicated that PPI and RBPI were more efficient at debasing particle size of nanoemulsions (pH 7.0) when they were at lower protein concentrations (2% *w*/*v*).

Zeta potential (ZP) is another important indicator of nanoemulsions stability. Because the surface charge highly commands the interactions between droplets in nanoemulsions. All nanoemulsions were composed of negatively charged droplets at pH 7.0 in our work (Figure 1b). This was because the number of carboxyl groups, which had a negative charge, was greater than the number of positively charged amino groups on proteins. Maximum ZP absolute values of 34.07 ± 1.27, 28.77 ± 1.17, 32.93 ± 1.42, and 32.40 ± 1.15 mV were recorded for PPI, RBPI, SPI, and WPI-stabilized nanoemulsions, respectively (Figure 1b). When the protein concentrations were lower or higher, the absolute values of ZP were decreased. It might be because the aqueous phase ionic strength was different, resulting in electrostatic screening [31]. The ZP values of previous research results were from −2 to −43 mV [32,33]. The relatively high ZP absolute values of PPI-stabilized nanoemulsions demonstrated that its stability was superior to RBPI, SPI, and WPI due to an intense electrostatic repulsion acting between the same charged droplets [32].

### 3.2. Effects of Oil Phase Fraction on Nanoeumlsions MDD and ZP

To investigate the influence of oil phase fraction on MDD and ZP, nanoemulsions were produced using different oil phase fraction (Figure 2a). For PPI and WPI-stabilized nanoemulsions, as oil phase fraction increased from 1% to 3%, the particle size decreased from 249.80 ± 2.26 to 227.30 ± 7.53 nm and from 275.50 ± 1.87 to 245.80 ± 0.35 nm, respectively. For RBPI and SPI-stabilized nanoemulsions, as oil phase fraction increased from 1% to 2% resulted in a decrease in droplet size from 278.70 ± 1.31 to 251.53 ± 3.11 nm and from 243.57 ± 0.97 to 233.40 ± 0.60 nm, respectively. However, as oil phase fraction further increased to 20%, the MDD of all nanoemulsions began to gradually increase (Figure 2a). Guo and Mu (2011) got similar results in the manufacture of corn oil nanoemulsions using sweet potato protein for emulsifier [34]. The initial reduction in MDD of nanoemulsions could be due to the reduced proportions of unabsorbed protein as the oil phase fraction increased, and thus a decrease in protein molecules aggregation. The subsequent higher oil phase fraction resulted in higher MDD. The phenomenon might be caused by the deficient mass of protein molecules available to overlay the oil droplets, which led to the enhancement of oil droplet coalescence [35].

Figure 2b depicts the effect of oil phase fraction on nanoemulsions ZP. The negative ZP was ranged from −35.23 to −25.10 mV when oil phase fraction increased from 1% to 20%. In case of oil phase fraction over 3% (for PPI and WPI) or 2% (for RBPI and SPI), ZP absolute value decreased with an increased oil phase fraction. The trend indicated that the electrostatic repulsion between droplets of nanoemulsions was increasingly less able to resist particle coalescence and flocculation. With a further increase in oil phase fraction, the absolute value of ZP decreased and the mean droplet diameter increased more significantly (*p* < 0.05) (Figure 2a). Therefore, the stability of protein-stabilized nanoemulsions decreased.

### 3.3. Effects of Ultrasonic Power and Time on Nanoemulsions MDD and ZP

As described above, the stable nanoemulsions preparation was concerned with the category and content of emulsifier, as well as oil phase fraction applied to droplets. But beyond that, in order to attain a high dispersion of oil droplets in the continuous phase, the high energy input is necessary to destroy the oil–water interface [32]. The determination of optimal ultrasonic power and time is very important for an industrial production of ultrasonic-mediated nanoemulsions. It can reduce energy wastage and cost of manufacture [36].

Increasing the ultrasonic power from 100 to 500 W led to a significant decrease in MDD (*p* < 0.05) (Figure 3a) and an increase in ZP absolute value (Figure 3b) for PPI, RBPI, SPI, and WPI-stabilized nanoemulsions. The results were similar to those obtained by Abbas et al. (2014) [32]. This was because as ultrasonic power increased, there was a concomitant increase in the applied sound pressure amplitude, thereby increasing cavitation intensity [37]. Then the energy consumption rate of system increased similarly, and the dispersion of one phase to another could be accelerated effectually [38]. Since the energy input was needed to be setup at a suitable condition to obtain a desired particle size, the ultrasonic power was setup at a maximum of 500 W for the next experiments.

The ultrasonic time represents the processing time of ultrasonic probe in nanoemulsions preparation. Next, nanoemulsions were prepared by varying ultrasonic time, at settled imposed power of 500 W. Effects of different sonication time (5–25 min) on MDD and ZP are in Figure 4. As the ultrasonic time increased from 5 to 25 min, MDD decreased and ZP absolute value increased for four proteins-stabilized nanoemulsions. Ultrasonic time of 20 min was discovered to be optimal, because there was no significant decrease in MDD over time (*p* > 0.05). In addition, prolonged sonication should be avoided as it might degrade the active ingredients of the product [32]. Therefore, subsequent experiments were performed at 20 min.

Under the optimum ultrasonic preparation conditions, the minimum particle size of PPI, RBPI, SPI, and WPI-stabilized nanoemulsions was 223.77 ± 1.79, 244.97 ± 1.66, 225.40 ± 2.18, 241.03 ± 2.50 nm, respectively. And the narrow distributions of the four proteins-stabilized nanoemulsions were revealed by the small PDI values (<0.3) (data not shown). PPI and SPI-stabilized nanoemulsions had smaller MDD than RBPI and WPI-stabilized nanoemulsions.

### 3.4. Apparent Viscosity of Nanoemulsions

The rheology of emulsion-based food systems is important because it influences their processing, functional properties and sensory attributes. The viscosities of four different protein-stabilized nanoemulsions are shown in Figure 5. For PPI, SPI and WPI-stabilized nanoemulsions, the slight shear thinning behavior was found when the shear rates were lower than 1 s^−1^; beyond this point, the viscosities no longer changed significantly as shear rate changed. For RBPI-stabilized nanoemulsions, the slight shear thinning behavior was found at shear rates below 10 s^−1^. This behavior was concerned with the aggregation of emulsions droplets, when the shear rate was sufficient to conquer Brownian motion, the nanoemulsions droplets were increasingly ordered within the flow field, offering less flow resistance and thus decreasing viscosity [39]. At high shear rate no significant viscosity changes were observed for four different protein-stabilized nanoemulsions, with behavior like that of high-shear Newtonian fluid [40]. However, the RBPI-stabilized nanoemulsions presented marginally higher viscosity and greater shear thinning than PPI, SPI and WPI-stabilized nanoemulsions at low shear rate values. This indicated that the structure of RBPI was destroyed and the protein molecules were aggregated during shearing process, which resulted in the larger size of RBPI-stabilized nanoemulsions than other three protein-stabilized nanoemulsions. And the results were further confirmed by confocal laser scanning micrographs of nanoemulsions (Figure 6).

### 3.5. Microstructure of Nanoemulsions

The microstructure of nanoemulsions was evaluated using confocal laser scanning microscopy (Figure 6). All nanoemulsions stabilized by four different proteins presented quite exquisite and uniform distributed droplets. And no intense interconnected oil droplets and protein networks were observed, which suggested that these proteins were effective emulsifier for stabilizing nanoemulsions. Luo et al. (2017) also viewed fairly exquisite droplets of dual-channel microfluidizer-mediated nanoemulsions, which were stabilized by WPI as emulsifier [41]. However, RBPI-stabilized nanoemulsions showed marginally larger oil droplets in comparison with PPI, SPI and WPI-stabilized nanoemulsions. The little increase of droplet size in RBPI-stabilized nanoemulsions might be the droplets coalescence phenomena, because RBPI-stabilized nanoemulsions had larger droplet size and lower ZP value (Figure 4). The higher viscosity of RBPI-stabilized nanoemulsions made droplets more likely to aggregate (Figure 5).

### 3.6. Storage Stability of Nanoemulsions

The effects of storage time on PPI, RBPI, SPI, and WPI-stabilized nanoemulsions physical and oxidative stability were determined (Figure 7). The nanoemulsions were stored under the condition of 4 °C for 4 weeks.

#### 3.6.1. Physical Stability

Physical stability of nanoemulsions is an essential factor to decide their suitability for application in food industries. The PPI, SPI, and WPI-stabilized nanoemulsions presented favorable resistance to phase separation during storage. Especially, PPI formed the most physically stable nanoemulsions, with no significant variation in particle size over time (*p* > 0.05) (Figure 7a). The MDD changed from 223.77 ± 1.79 to 226.63 ± 2.06 nm in PPI-stabilized nanoemulsions. This might be ascribed to the fact that the emulsion droplets were fairly stable to gravity separation, forming a stout interfacial film to resist cracking [42,43]. However, a great MDD variation with time could be observed for RBPI-stabilized nanoemulsions. The MDD of RBPI-stabilized nanoemulsions was greater than 1000 nm after 3 weeks. And the PDI values of RBPI-stabilized nanoemulsions were greater than 0.3 after 3 weeks (data not shown). The results indicated that some formation of droplet aggregation had occurred, such as flocculation or coalescence, resulting in poor physical stability during storage. From the above experiments we could arrive at the conclusion that PPI, SPI, and WPI-stabilized nanoemulsions revealed appropriate stability at 4 °C for 4 weeks.

#### 3.6.2. Oxidative Stability

The oxidative stability of PPI, RBPI, SPI, and WPI-stabilized nanoemulsions was analyzed by quantifying malondialdehyde (Figure 7b). At the initial phase of storage, all nanoemulsions had an increase in TBARS values, suggesting that lipid oxidation had occurred. And the increase in TBARS values of the RBPI-stabilized nanoemulsions during storage was very significant (*p* < 0.05), which might be concerned with the occurrence of droplet coalescence. The coalescence might result in a closer lipid phase, thus promoting the movement of the pro-oxidants in the oil phase [44]. The TBARS values of the nanoemulsions stabilized by PPI had been the lowest of the four proteins during storage. This could be because that the far higher negative charge of PPI-stabilized nanoemulsions (Figure 4b) was more intense in combination with transition metal irons, resulting in that the nanoemulsion was more stable [45]. The maximum malondialdehyde content of the four proteins-stabilized nanoemulsions during storage was 1.35 ± 0.03 mg/kg oil. However, the quantities of oxidation products produced during 4 weeks of storage were below the maximum endurable standard (1–2 mg of malondialdehyde/kg oil), indicating no effect on the quality of the products [43].

### 3.7. Characteristics of Proteins

#### 3.7.1. Secondary Structure

When the protein spread out at the two-phase interface, its specific distribution in the oil-water phase is related to its secondary structure. The ratios of α-helix, β-sheet, β-turn, and random coil for these four proteins were listed in Table 1. For PPI, the contents of ordered structure (α-helix + β-sheet) were relatively low, and the contents of unordered structure (β-turn + random coil) were relatively high, while the unordered structure contents of RBPI were relatively smaller. It had shown that an increasingly disordered structure can improve adsorption at the oil-water interface, enhancing emulsifying properties [46]. It could also be seen that PPI-stabilized nanoemulsions had the smallest particle size (Figure 4a). And the PPI-stabilized nanoemulsions were pretty stable to droplet aggregation during storage, while the stability of RBPI-stabilized nanoemulsions was relatively poor (Figure 7a).

#### 3.7.2. Interfacial Characteristics

The formation and stability of nanoemulsions are related to the nature of the emulsifiers, which is dependent upon the characteristics of the interface layer surrounding the oil droplets. Thus, interfacial tensions were measured at different protein concentrations (Figure 8). The lower the interfacial tension of the protein indicates the smaller the particle size of the nanoemulsion formation, because less destructive forces are required to break the droplets during preparation [47]. The initial IT between corn oil and distilled water was 21.3 ± 0.2 mN/m. The IT of four proteins appeared to decline gradually as concentrations increased, indicating that the protein molecules adsorbed to the oil/water interface and shielded the unfavorable molecular interactions between oil and water [48]. Compared with RBPI, PPI was more effective at lowering IT with small concentration. This might be because the PPI-stabilized nanoemulsions had higher ZP values (Figure 4b), resulting in greater repulsive forces between protein molecules. The repulsive forces caused the protein molecules to unfold and better adsorb at the oil/water interface [23]. This might explain that PPI based nanoemulsions had higher stability among four nanoemulsions (Figure 7).

### 3.8. Correlation Analysis

In this study, the data of nanoemulsions prepared under optimal conditions of four proteins were selected as typical for analyzing the correlation between protein secondary structure, MDD, and ZP. As shown in Table 2, the correlation analysis showed that α-helix (*p* < 0.01) and β-turn (*p* < 0.05) of protein were significantly correlated with the MDD of nanoemulsions, the random coil (*p* < 0.05) was significantly related to the ZP of nanoemulsions. It was obvious that differences in protein structure could affect characteristics of nanoemulsions. The results further confirmed that the disordered structure of protein was conducive to the formation and stabilization of the nanoemulsions.

## 4. Conclusions

PPI and RBPI could produce nanoemulsions with fairly uniform small droplets (<300 nm) as efficiently as WPI and SPI. The nanoemulsions formed by PPI had a smaller MDD and a higher ZP as compared to SPI, RBPI and WPI. Furthermore, PPI, SPI and WPI-stabilized nanoemulsions exhibited substantial stability during 4 weeks storage at 4 °C. RBPI-stabilized nanoemulsions were less stable, and the mean droplet diameter had exceeded 1000 nm during the third week of storage. The results indicated that PPI was an excellent emulsifier for preparing nanoemulsions. The more disordered structure and lower IT of protein was favorable to form stable nanoemulsions. The correlation analysis revealed that the MDD of nanoemulsions was affected by α-helix (*p* < 0.01) and β-turn (*p* < 0.05) of protein, and the ZP of nanoemulsions was affected by random coil of protein (*p* < 0.05). The research provided novel insights regarding the relationship between protein structure, the formation and stabilization of protein-stabilized nanoemulsions.

## Figures and Tables

**Figure 1 nanomaterials-09-00025-f001:**
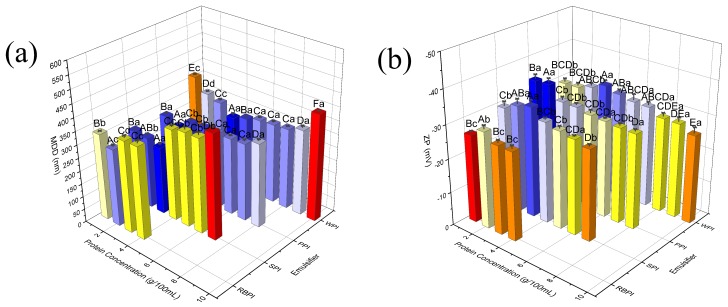
Effects of protein concentrations on MDD (**a**) and ZP (**b**) of PPI, RBPI, SPI, and WPI-stabilized nanoemulsions. Samples marked by uppercase letters (A–F) indicate significant difference (*p* < 0.05) relative to different concentrations of the same protein. Samples marked by lowercase letters (a–d) indicate significant difference (*p* < 0.05) when compared between different proteins of the same concentration. The PPI, RBPI, SPI, and WPI-stabilized nanoemulsions were fabricated under the circumstance of corn oil (5% *v*/*v*), ultrasonic power 400 W and ultrasonic time 15 min.

**Figure 2 nanomaterials-09-00025-f002:**
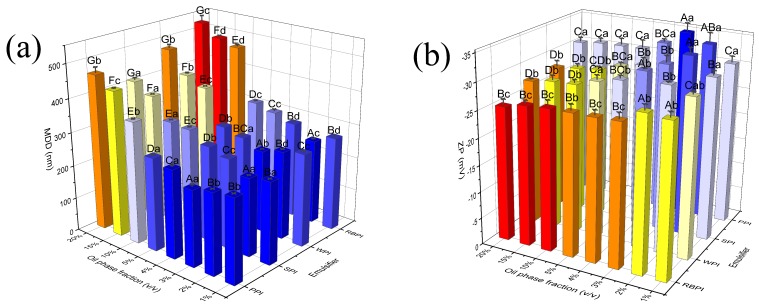
Effects of oil phase fraction on mean droplet diameter (MDD) (**a**) and zeta potential (ZP) (**b**) of peanut protein isolate (PPI), rice bran protein isolate (RBPI), soybean protein isolate (SPI), and whey protein isolate (WPI)-stabilized nanoemulsions. Samples marked by uppercase letters (A–G) indicate significant difference (*p* < 0.05) relative to different oil phase fractions. Samples marked by lowercase letters (a–d) indicate significant difference (*p* < 0.05) when compared between different proteins of the same oil phase fraction. The PPI, RBPI, SPI, and WPI-stabilized nanoemulsions were fabricated under the circumstance of the different protein concentrations of PPI (2% *w*/*v*), RBPI (2% *w*/*v*), SPI (3% *w*/*v*), and WPI (4% *w*/*v*), ultrasonic power 400 W and ultrasonic time 15 min.

**Figure 3 nanomaterials-09-00025-f003:**
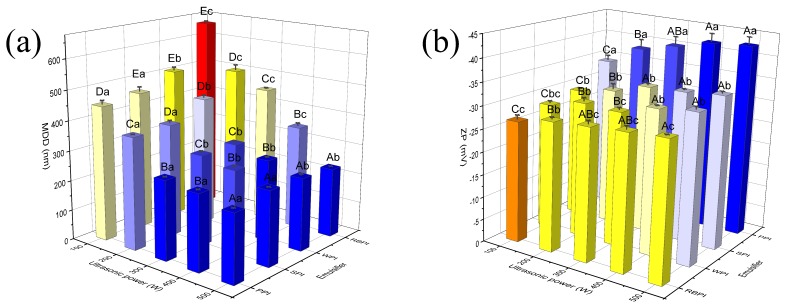
Effects of ultrasonic power on MDD (**a**) and ZP (**b**) of PPI, RBPI, SPI, and WPI-stabilized nanoemulsions. Samples marked by uppercase letters (A–E) indicate significant difference (*p* < 0.05) when compared between different ultrasonic power. Samples marked by lowercase letters (a–c) indicate significant difference (*p* < 0.05) when compared between different proteins of the same ultrasonic power. The PPI, RBPI, SPI, and WPI-stabilized nanoemulsions were fabricated under the circumstance of the different protein concentrations of PPI (2% *w*/*v*), RBPI (2% *w*/*v*), SPI (3% *w*/*v*), and WPI (4% *w*/*v*), corn oil (3%, 2%, 2%, 3% *v*/*v*, respectively) and ultrasonic time 15 min.

**Figure 4 nanomaterials-09-00025-f004:**
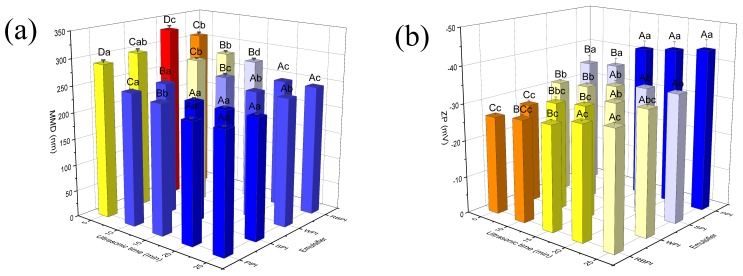
Effects of ultrasonic time on MDD (**a**) and ZP (**b**) of PPI, RBPI, SPI, and WPI-stabilized nanoemulsions. Samples marked by uppercase letters (A–D) indicate significant difference (*p* < 0.05) when compared between different ultrasonic time. Samples marked by lowercase letters (a–d) indicate significant difference (*p* < 0.05) when compared between different proteins of the same ultrasonic time. The PPI, RBPI, SPI and WPI-stabilized nanoemulsions were fabricated under the circumstance of the different protein concentrations of PPI (2% *w*/*v*), RBPI (2% *w*/*v*), SPI (3% *w*/*v*) and WPI (4% *w*/*v*), corn oil (3%, 2%, 2%, 3% *v*/*v*, respectively) and ultrasonic power 500W.

**Figure 5 nanomaterials-09-00025-f005:**
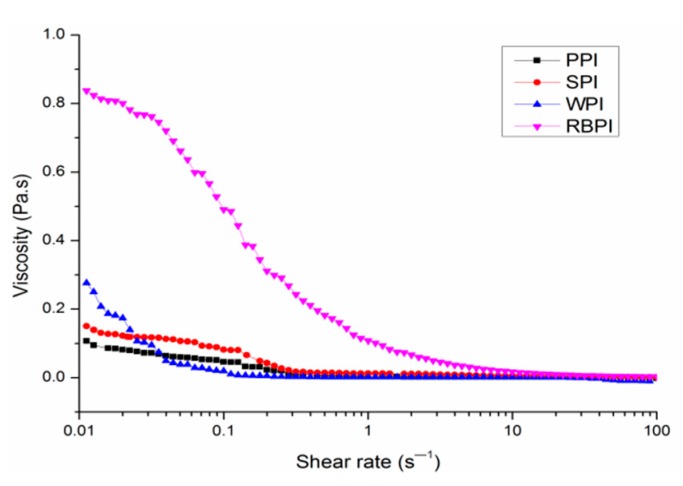
Rheological properties of PPI, SPI, WPI, and RBPI-stabilized nanoemulsions. The PPI, RBPI, SPI, and WPI-stabilized nanoemulsions were fabricated under the circumstance of the different protein concentrations of PPI (2% *w*/*v*), RBPI (2% *w*/*v*), SPI (3% *w*/*v*), and WPI (4% *w*/*v*), corn oil (3%, 2%, 2%, 3% *v*/*v*, respectively), ultrasonic power 500 W and ultrasonic time 20 min.

**Figure 6 nanomaterials-09-00025-f006:**
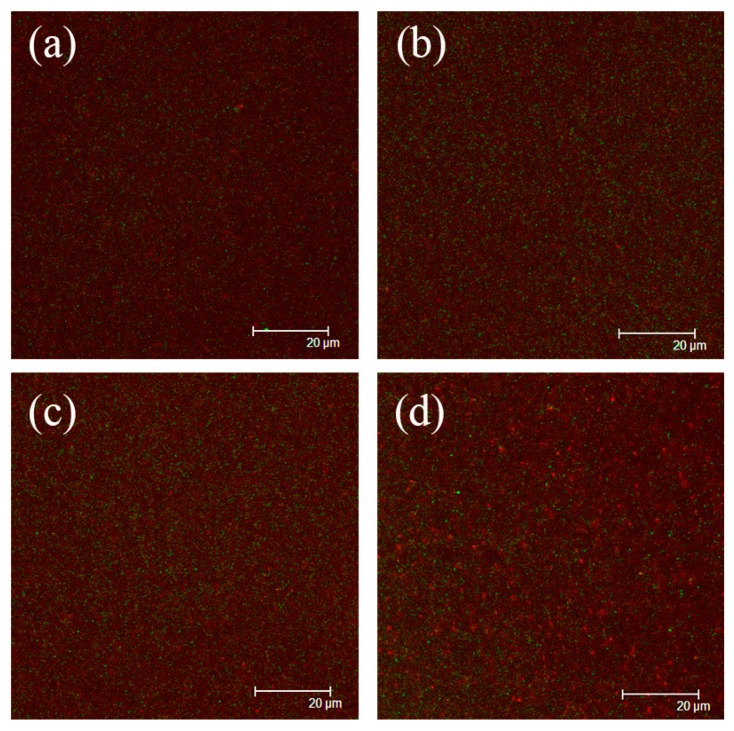
Confocal laser scanning micrographs of PPI (**a**), SPI (**b**), WPI (**c**), and RBPI (**d**)-stabilized corn oil nanoemulsions. The PPI, RBPI, SPI, and WPI-stabilized nanoemulsions were fabricated under the circumstance of the different protein concentrations of PPI (2% *w*/*v*), RBPI (2% *w*/*v*), SPI (3% *w*/*v*), and WPI (4% *w*/*v*), corn oil (3%, 2%, 2%, 3% *v*/*v*, respectively), ultrasonic power 500 W and ultrasonic time 20 min.

**Figure 7 nanomaterials-09-00025-f007:**
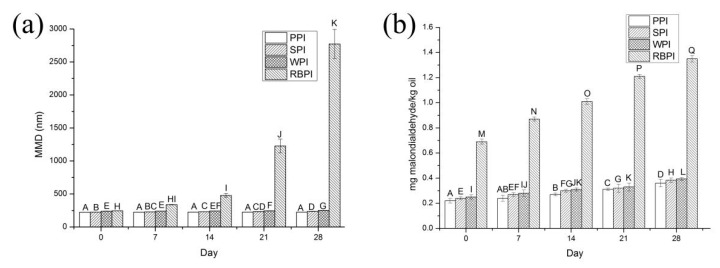
Influence of storage time on MDD (**a**) and thiobarbituric acid-reactive substances (TBARS) values (**b**) of PPI, SPI, WPI, and RBPI-stabilized nanoemulsions. Samples designated with different uppercase letters (A), (B–D), (E–G), and (H–K) indicate significant difference (*p* < 0.05) when compared with different storage time on MDD of PPI, SPI, WPI, and RBPI -stabilized nanoemulsions, respectively. Samples designated with different upper case letters (A–D), (E–H), (I–L) and (M–Q) indicate significant difference (*p* < 0.05) when compared between different storage time on TBARS values of PPI, SPI, WPI, and RBPI-stabilized nanoemulsions, respectively. The PPI, RBPI, SPI, and WPI-stabilized nanoemulsions were fabricated under the circumstance of the different protein concentrations of PPI (2% *w*/*v*), RBPI (2% *w*/*v*), SPI (3% *w*/*v*), and WPI (4% *w*/*v*), corn oil (3%, 2%, 2%, 3% *v*/*v*, respectively), ultrasonic power 500 W and ultrasonic time 20 min.

**Figure 8 nanomaterials-09-00025-f008:**
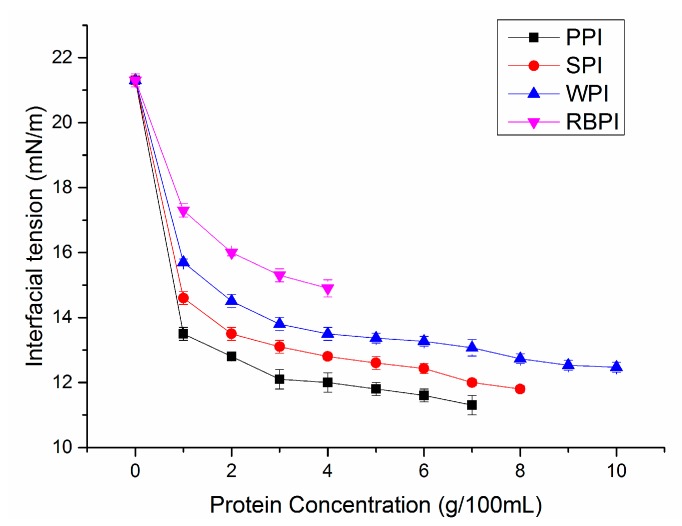
Influence of different concentrations of PPI, SPI, WPI, and RBPI protein dispersions on interfacial tension against corn oil.

**Table 1 nanomaterials-09-00025-t001:** Secondary structure contents of PPI, RBPI, SPI, and WPI protein dispersions.

Sample	α-Helix (%)	β-Sheet (%)	β-Turn (%)	Random Coil (%)
PPI	12.01 ± 0.03 ^a^	30.17 ± 0.03 ^a^	41.76 ± 0.03 ^d^	16.06 ± 0.03 ^d^
RBPI	16.90 ± 0.01 ^d^	34.63 ± 0.03 ^c^	34.64 ± 0.02 ^b^	13.83 ± 0.01 ^a^
SPI	12.84 ± 0.02 ^b^	31.44 ± 0.03 ^b^	40.99 ± 0.04 ^c^	14.72 ± 0.02 ^c^
WPI	15.68 ± 0.02 ^c^	35.65 ± 0.02 ^d^	34.11 ± 0.04 ^a^	14.56 ± 0.03 ^b^

Means with dissimilar lower case letters (a, b, c, and d) indicate significance (*p* < 0.05).

**Table 2 nanomaterials-09-00025-t002:** Correlation analysis between the structure of protein and properties of protein-stabilized nanoemulsions.

	α-Helix	β-Sheet	β-Turn	Random	MDD	ZP
α-Helix	1					
β-Sheet	0.928	1				
β-Turn	−0.961 *	−0.986 *	1			
Random	−0.872	−0.784	0.767	1		
MDD	0.994 **	0.942	−0.979 *	−0.817	1	
ZP	0.824	0.808	−0.761	−0.980 *	0.773	1

Value indicated by an asterisk is significant at the 0.05 level (*p* < 0.05). Value represented by two asterisks is significant at the 0.01 level (*p* < 0.01).

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
