# Peer review of "Physicochemical Properties and Storage Stability of Food Protein-Stabilized Nanoemulsions"

_nanomaterials, 2018, doi:10.3390/nano9010025_

Reviewer 1 Report

Physicochemical properties and stability of protein-stabilized corn oil-in-water nanoemulsions prepared by ultrasonic method were studied in this work. The results obtained are of interest and many experiments have been performed. However, I think that the manuscript should be revised and improved before its publication, taking into account the following points:

1) Please include the oil used (corn oil) in the Abstract section.

2) Page 3, Section 2.2. More details are needed on nanoemulsion formulation and preparation. What kind of water was used for protein solutions? Which probe (tip diameter) was used in the ultrasonic processor for nanoemulsion preparation? Ultrasonic power was specified (100-500 W) in the manuscript, but not the amplitude used. Have the ultrasounds been applied continuously or in pulses for nanoemulsion preparation?

3) Page 3, lines 96-97. This sentence is unclear, please rewrite it.

4) Page 4, line 127. “Du Nouy” instead of “DU Nouy”.

5) There is no information on pH, a very important parameter when working with proteins that can affect their structure and functional properties, especially if it is close to their isoelectric point. What is the pH of the protein solutions and nanoemulsions (7.0, as indicated in page 5, line 159?)? Has a buffer solution been added to the feed to avoid pH changes and obtain comparable results? Has the pH variation been measured after the formulation and ultrasonic process?

6) The authors only indicate the mean droplet diameter (MDD) of the emulsions, but there is also an important parameter that can be determined using the Zetasizer Nano ZS equipment: the polydispersity index (PDI), which indicates the width of a particle size distribution measured by light scattering techniques. A PDI < 0.3 is usually required to obtain a stable nanoemulsion. Do the authors have any data on this parameter for their formulated nanoemulsions?

7) Results and Discussion section should be revised to include several missing information in the original manuscript:

- Please be careful when explaining zeta potential (ZP) results: always refer to the absolute values of ZP (i.e. “Maximum ZP absolute values” (page 5, line 161), “the values of ZP were increased” or “the absolute values of ZP were decreased” (page 5, lines 163-164), “a decrease in ZP” (page 6, line 211)).

- Please indicate the composition of nanoemulsions in the figure captions: oil content in Figure 1, emulsifier content in Figure 2, and oil and emulsifier content (and ultrasonic time or ultrasonic power) in Figures 3-8 (I suppose that nanoemulsions in Figures 5-8 were formulated with an ultrasonic time of 20 minutes, as indicated on page 7, line 231).

8) Refs. [40]-[48] are mentioned in text but not included in the References section.

Author Response

Thank you for your comments and allowing us to address your concerns below. We have modified the manuscript accordingly and all changes were also highlighted (red) throughout the manuscript. The manuscript has been resubmitted and we look forward to your positive response.

1. Please include the oil used (corn oil) in the Abstract section.

Authors: Thank you for your kind suggestions. We have included the oil used (corn oil) in the Abstract section (Line 11) and highlighted (red).

2. Page 3, Section 2.2. More details are needed on nanoemulsion formulation and preparation. What kind of water was used for protein solutions? Which probe (tip diameter) was used in the ultrasonic processor for nanoemulsion preparation? Ultrasonic power was specified (100-500 W) in the manuscript, but not the amplitude used. Have the ultrasounds been applied continuously or in pulses for nanoemulsion preparation?

Authors: Thank you for your comment. The stock protein solutions were dissolved in 10 mM phosphate buffer (pH 7.0) and agitated well to ensure complete solubilization. The ultrasonic processor had been applied in pulses (on-time 2 s and off-time 3 s) for nanoemulsion preparation with a 0.636 cm diameter titanium. We have added “(10 mM phosphate buffer, pH 7.0)” after “aqueous phase” (Line 84), “with a 0.636 cm diameter titanium probe” after “ultrasonic processor (Ningbo Xinzhi Biotechnology Co. Ltd., Ningbo, China) ” (Line 90), changed “ultrasonic power (100-500 W)” in “ultrasonic power (100-500 W, corresponding to the amplitude of 10%-50%)” (Line 91), and changed “time (5-25 min)” in “time (5-25 min, pulse duration of on-time 2 s and off-time 3 s)” (Line 91) and highlighted (red) in Section 2.2.

3. Page 3, lines 96-97. This sentence is unclear, please rewrite it.

Authors: Thank you for your kind suggestions. We have rewrited this sentence that “The particle size results were reported as the Z-average mean diameter calculated from the particle size distribution. To avoid multiple light scattering effects, nanoemulsions were diluted 100-fold with 10 mM phosphate buffer (pH 7.0) and agitated well at 25 °C. The refractive index values used for oil (corn oil) and dispersant (phosphate buffer) were 1.47 and 1.33, respectively.” (Lines 96-100)

4. Page 4, line 127. “Du Nouy” instead of “DU Nouy”.

Authors: Thank you for your kind suggestions and “Du Nouy” was instead of “DU Nouy”. (Line 136)

5. There is no information on pH, a very important parameter when working with proteins that can affect their structure and functional properties, especially if it is close to their isoelectric point. What is the pH of the protein solutions and nanoemulsions (7.0, as indicated in page 5, line 159?)? Has a buffer solution been added to the feed to avoid pH changes and obtain comparable results? Has the pH variation been measured after the formulation and ultrasonic process? 

Authors: Thank you for your kind suggestions. The stock PPI (1-8% w/v), RBPI (1-4% w/v), SPI (1-7% w/v) or WPI (1-10% w/v) were dissolved in 10 mM phosphate buffer (pH 7.0) to avoid pH changes. The sections were supplemented on Line 84 and Line 161. In this experiment, the pH of the protein solutions and nanoemulsions was 7.0, and the pH of the nanoemulsions didn’t change after the formulation and ultrasonic process.

6. The authors only indicate the mean droplet diameter (MDD) of the emulsions, but there is also an important parameter that can be determined using the Zetasizer Nano ZS equipment: the polydispersity index (PDI), which indicates the width of a particle size distribution measured by light scattering techniques. A PDI < 0.3 is usually required to obtain a stable nanoemulsion. Do the authors have any data on this parameter for their formulated nanoemulsions?

Authors: Thank you for your kind suggestions. We determined PDI for formulated nanoemulsions. The sections were supplemented on Lines 248-249 and Lines 313-314. Data were as follows:

Table 1. Changes of polydispersity index (PDI) with protein concentration

Protein Concentration (g/100mL)

PDI

PPI

SPI

WPI

RBPI

1

0.212±0.015

0.230±0.011

0.221±0.016

0.233±0.011

2

0.177±0.008

0.207±0.004

0.195±0.011

0.216±0.010

3

0.183±0.017

0.184±0.004

0.199±0.008

0.240±0.018

4

0.180±0.011

0.196±0.013

0.190±0.006

0.265±0.011

5

0.199±0.009

0.193±0.009

0.213±0.008

——

6

0.201±0.016

0.209±0.011

0.211±0.014

——

7

0.211±0.009

0.205±0.010

0.221±0.010

——

8

0.218±0.007

——

0.224±0.007

——

9

——

——

0.231±0.011

——

10

——

——

0.240±0.009

——

Table 2. Changes of polydispersity index (PDI) with oil phase fraction 

Oil phase fraction (v/v)

PDI

PPI

SPI

WPI

RBPI

1

0.205±0.012

0.184±0.016

0.224±0.016

0.231±0.012

2

0.177±0.016

0.180±0.010

0.193±0.011

0.207±0.008

3

0.163±0.004

0.197±0.014

0.187±0.011

0.209±0.009

4

0.165±0.008

0.203±0.019

0.203±0.007

0.226±0.015

5

0.201±0.007

0.196±0.009

0.198±0.004

0.217±0.004

10

0.188±0.013

0.212±0.011

0.196±0.006

0.237±0.009

15

0.216±0.010

0.209±0.015

0.224±0.013

0.231±0.015

20

0.210±0.007

0.204±0.010

0.258±0.017

0.248±0.010

Table 3. Changes of polydispersity index (PDI) with ultrasonic power 

Ultrasonic power (W)

PDI

PPI

SPI

WPI

RBPI

100

0.225±0.008

0.247±0.016

0.269±0.012

0.288±0.009

200

0.207±0.011

0.210±0.010

0.205±0.015

0.244±0.014

300

0.184±0.009

0.194±0.013

0.203±0.006

0.216±0.006

400

0.181±0.006

0.178±0.006

0.188±0.010

0.197±0.009

500

0.143±0.006

0.159±0.008

0.164±0.004

0.189±0.007

Table 4. Changes of polydispersity index (PDI) with ultrasonic time

Ultrasonic time (min)

PDI

PPI

SPI

WPI

RBPI

5

0.209±0.011

0.229±0.019

0.237±0.017

0.267±0.014

10

0.168±0.006

0.194±0.013

0.191±0.013

0.208±0.015

15

0.131±0.010

0.162±0.011

0.179±0.008

0.204±0.011

20

0.101±0.009

0.118±0.009

0.144±0.010

0.185±0.012

25

0.094±0.005

0.113±0.008

0.136±0.007

0.176±0.007

Table 5. Changes of polydispersity index (PDI) with storage time

Day

PDI

PPI

SPI

WPI

RBPI

0

0.101±0.005

0.118±0.008

0.144±0.007

0.185±0.007

7

0.107±0.011

0.122±0.006

0.147±0.007

0.237±0.016

14

0.122±0.009

0.145±0.011

0.154±0.016

0.286±0.019

21

0.139±0.013

0.147±0.009

0.162±0.013

0.397±0.023

28

0.163±0.010

0.181±0.015

0.190±0.010

0.559±0.020

 7. Results and Discussion section should be revised to include several missing information in the original manuscript:

- Please be careful when explaining zeta potential (ZP) results: always refer to the absolute values of ZP (i.e. “Maximum ZP absolute values” (page 5, line 161), “the values of ZP were increased” or “the absolute values of ZP were decreased” (page 5, lines 163-164), “a decrease in ZP” (page 6, line 211)).

- Please indicate the composition of nanoemulsions in the figure captions: oil content in Figure 1, emulsifier content in Figure 2, and oil and emulsifier content (and ultrasonic time or ultrasonic power) in Figures 3-8 (I suppose that nanoemulsions in Figures 5-8 were formulated with an ultrasonic time of 20 minutes, as indicated on page 7, line 231).

Authors: Thank you for your kind suggestions.

- We have changed “the absolute values of ZP, ZP or the values of ZP” in “ZP absolute values or the absolute values of ZP”. The changes were also highlighted (red) in the manuscript (Lines 167-168, 172, 200, 222 and 241). 

- We have indicated the composition of nanoemulsions in the figure 1-7 captions  (Lines 180-182, 210-213, 235-237, 256-258, 278-281, 296-299 and 341-344) and highlighted (red).

8. Refs. [40]-[48] are mentioned in text but not included in the References section.

Authors: Thank you for your kind suggestions. We have added Refs. [40]-[48] in the References section (Lines 507-530) and highlighted (red). 

Reviewer 2 Report

The manuscript entitled “Physicochemical Properties and Storage Stability of Food Proteins-Stabilized Nanoemulsions Prepared  Via Ultrasound Method” by  Yangyang Li, Hua Jin, Xiaotong Sun, Jingying Sun, Chang Liu, Chunhong Liu and Jing Xu deals with the production of corn oil nanoemulsions stabilized by different proteins. The manuscript is interesting and a great number of works has been carried out but I think that a revision is requested before its publication.

 First of all, I suggest the Authors change the manuscript title because the improvements derived by the use of ultrasounds are not highlighted in the manuscript. There are proofs made at different sonication time and power but they are not compared with the results and the effects of other mixing methods.

Regarding the experimental description, there is a general lack of details.  Some suggestions are reported as follows:

- Please give more details on the light scattering and zeta potential measurements. I suppose your samples were very turbid. Did you dilute your samples for determining MDD and ZP?

- Also for rheology measurements, please give more details. What kind of geometry cell have you used? The temperature?

 Concerning the results presentation, I suggest the authors to give an in-depth reading to their manuscript to avoid trivial and inappropriate expressions.

For example, the concept of “sectors” recalled at page 4 sounds unclear to me. What do the authors mean with Sectors? Please rephrase lines 143-150 on page 4 and explain better.

What the author mean with the “dwell time of emulsion” ?(page 7 line 224)

 Finally, in the results, there are important details missing. 

Which was the oil content of the emulsions represented in Figure 1?

Which was the emulsifier content of the emulsions represented in Figure 2?

And the oil and emulsifier contents of emulsions in Figures 3 and 4 and 5?

For the samples reported in Fig 5 (showing the apparent viscosity of the emulsions) the combination of time and power for the sonication mixing was the same for all of them?

Details on the sample composition are missing also for figure 6 and 7.

Please provide the sample composition in the in the main text or in the figure captions.

 Author Response

Thank you for your comments and allowing us to address your concerns below. We have modified the manuscript accordingly and all changes were also highlighted (red) throughout the manuscript. The manuscript has been resubmitted and we look forward to your positive response.

1. First of all, I suggest the Authors change the manuscript title because the improvements derived by the use of ultrasounds are not highlighted in the manuscript. There are proofs made at different sonication time and power but they are not compared with the results and the effects of other mixing methods.

Authors: Thank you for your kind suggestions. We have changed the manuscript title that “Physicochemical Properties and Storage Stability of Food Proteins-Stabilized Nanoemulsions” and highlighted (red).

2. Regarding the experimental description, there is a general lack of details.  Some suggestions are reported as follows:

- Please give more details on the light scattering and zeta potential measurements. I suppose your samples were very turbid. Did you dilute your samples for determining MDD and ZP?

- Also for rheology measurements, please give more details. What kind of geometry cell have you used? The temperature?

Authors: Thank you for your kind suggestions.

- To avoid multiple light scattering effects, nanoemulsions were diluted 100-fold with 10 mM phosphate buffer (pH 7.0) and agitated well at 25 °C. We have provided more details on the light scattering and zeta potential measurements (Lines 96-100 and 102-104) and highlighted (red).

- The sections of rheology measurements were supplemented on Lines 106-108.

3. Concerning the results presentation, I suggest the authors to give an in-depth reading to their manuscript to avoid trivial and inappropriate expressions.

For example, the concept of “sectors” recalled at page 4 sounds unclear to me. What do the authors mean with Sectors? Please rephrase lines 143-150 on page 4 and explain better.

What the author mean with the “dwell time of emulsion” ?(page 7 line 224)

Authors: Thank you for your kind suggestions. In order to make this part more clearly for readers, we rewrited it (Lines 147-155) and highlighted (red) in the manuscript.

The “dwell time of emulsion” means that “processing time of ultrasonic probe in nanoemulsions preparation”. We rewrited the sentences (Lines 238-239). 

4. Finally, in the results, there are important details missing. 

Which was the oil content of the emulsions represented in Figure 1?

Which was the emulsifier content of the emulsions represented in Figure 2?

And the oil and emulsifier contents of emulsions in Figures 3 and 4 and 5?

For the samples reported in Fig 5 (showing the apparent viscosity of the emulsions) the combination of time and power for the sonication mixing was the same for all of them?

Details on the sample composition are missing also for figure 6 and 7.

Please provide the sample composition in the in the main text or in the figure captions.

Authors: Thank you for your kind suggestions. We have indicated the sample composition in the figure 1-7 captions (Lines 180-182, 210-213, 235-237, 256-258, 278-281, 296-299 and 341-344) and highlighted (red). 
